# Lifting More Than Two Loads Compromises the Magnitude of the Load–Velocity Relationship Variables: Evidence in Two Variants of the Prone Bench Pull Exercise

Sergio Miras-Moreno [1], Amador García-Ramos [1,2], John F. T. Fernandes [3] and Alejandro Pérez-Castilla [4,5,*]

1 Department of Physical Education and Sport, Faculty of Sport Sciences, University of Granada, 18010 Granada, Spain
2 Department of Sports Sciences and Physical Conditioning, Faculty of Education, Universidad Católica de la Santísima Concepción, Concepción 2850, Chile
3 School of Sport and Health Sciences, Cardiff Metropolitan University, Cardiff CF23 6XD, UK
4 Department of Education, Faculty of Education Sciences, University of Almería, 04120 Almería, Spain
5 SPORT Research Group (CTS-1024), CERNEP Research Center, University of Almería, 04120 Almería, Spain
* Correspondence: alexperez@ual.es

**Abstract:** This study aimed to compare and associate the magnitude of the load–velocity relationship variables between the multiple-point and two-point methods and between the concentric-only and eccentric–concentric prone bench pull (PBP) variants. Twenty-three resistance-trained males completed a preliminary session to determine the concentric-only PBP one-repetition maximum (1 RM) and two experimental sessions that only differed in the PBP variant evaluated. In each experimental session they performed three repetitions against the 14 kg load (L1), two repetitions against the 85% 1 RM load (L4), three repetitions against an equidistant intermediate light load (L2), two repetitions against an equidistant intermediate heavy load (L3), and 1–5 1 RM attempts. The load–velocity relationship variables (i.e., load–axis intercept, velocity–axis intercept, and area under the load–velocity relationship line) were obtained from the multiple-point (L1-L2-L3-L4) and two-point (L1-L4) methods. All load–velocity relationship variables presented greater magnitudes when obtained by the two-point method compared with the multiple-point method ($p < 0.001$, $ES_{range} = 0.17$–$0.43$), while the load–velocity relationship variables were comparable between both PBP variants ($p \geq 0.148$). In addition, the load–velocity relationship variables were highly correlated between both methods ($r_{range} = 0.972$–$0.995$) and PBP variants ($r_{range} = 0.798$–$0.909$). When assessing the load–velocity relationship variables, practitioners should prescribe only two loads, as this maximises the magnitudes of the variables and decreases fatigue.

**Keywords:** concentric-only; eccentric–concentric; multiple-point method; two-point method; velocity-based training

## 1. Introduction

Because of the widespread use of sports technology and its exciting practical applications, velocity-based resistance training has made significant progress in the field of strength and conditioning [1,2]. One of the main practical applications consists of recording the load–velocity (L-V) profile for testing and monitoring purposes [3]. For example, the L-V relationship has been recently proposed as a simpler approach to assess the maximal capacities of the muscles to produce force at low (load–axis intercept ($L_0$)) and high (velocity–axis intercept ($v_0$)) velocities as well as performing work at a maximal rate (area under the L-V relationship line ($A_{line}$)) [4]. Since force data are not necessary for the L-V modelling, this recommendation is mainly geared to the lower extrapolation needed from the experimental points to $v_0$ in exercises performed against gravity [4]. Emerging evidence has confirmed the high reliability, sensitivity, and validity of the L-V relationship variables

with respect to other traditional tests [4–7]. However, there is a dearth of literature available for practitioners on how to optimise the L-V relationship testing procedures.

The L-V relationship is commonly determined by applying a linear regression model to the data acquired from multiple external loads (i.e., *"multiple-point method"*) [4,7,8]. However, since the L-V relationship is approximately linear during multi-joint tasks [9], the addition of intermediate loads should not meaningfully improve the accuracy of the L-V relationship [10]. Accordingly, the same L-V relationship variables obtained from the multiple-point method can also be obtained by applying only two distant loads in the L-V modelling (i.e., the *"two-point method"*) [11]. To date, only Pérez-Castilla et al. [5] have demonstrated that the L-V relationship variables during the countermovement jump exercise can be determined using the two-point approach, which is extremely valid in comparison with the multiple-point method. Particularly, the fact that the two-point method is derived from the testing procedure based on the multiple-point method has a significant impact on its validity. More importantly, the two-point method can considerably reduce testing time and minimise the fatigue induced by the testing protocol and, consequently, this simplified procedure is a better representation of maximal neuromuscular capabilities [12]. However, currently there is a need to assess the feasibility of the two-point method applied in field conditions (i.e., with only two distant loads used during the testing procedure) and in other core exercises (e.g., prone bench pull (PBP)).

The PBP is likely the most-used exercise in strength and conditioning programs to develop upper-body pulling strength [13,14]. Moreover, pulling actions are of considerable importance for success in various sports disciplines (e.g., sailing and rowing) [15,16]. Typically, the subject starts the pulling phase with the barbell motionless and raised, such that their arms are straight beneath the bench (i.e., the *"concentric-only PBP variant"*) [6,13,14,17]. The PBP exercise is therefore characterised by a descending strength curve where maximum strength is produced at the start of the lift [13]. However, when multiple repetitions are performed within a PBP training set, the pulling phase may also be preceded by a downward movement (i.e., the *"eccentric–concentric PBP variant"*) [18]. It is plausible that this eccentric overload may improve the muscles' ability to produce force earlier in the PBP motion by reducing muscle slack (i.e., the delay between muscular contraction and recoil of the series elastic elements) [19]. However, to the best of our knowledge, no study has compared the L-V profile between the two variants of PBP exercise. Similarly, it is of interest to elucidate whether both PBP variants provide similar insight into maximal neuromuscular capacities.

To overcome the abovementioned limitations and gaps in the literature, we evaluated, on two separate occasions, the L-V relationship using different modelling procedures during observed PBP variants. Specifically, the aims of the study were (i) to compare and (ii) to associate the magnitude of the L-V relationship variables between the multiple-point and two-point methods and between the concentric-only and eccentric–concentric PBP variants in resistance-trained males. We hypothesised that the L-V relationship variables would be as follows: (i) greater for the two-point method compared with the multiple-point method [12], (ii) greater for the eccentric–concentric PBP variant compared with the concentric-only PBP variant [19], and (iii) highly correlated between methods and PBP variants [5,20]. The expected findings should provide valuable information to refine this novel velocity-based methodology proposed to assess maximal neuromuscular capacities from a more practical point of view.

## 2. Materials and Methods

### 2.1. Subjects

Twenty-three resistance-trained males (mean ± standard deviation [SD]: age = 25.0 ± 7.3 years (range: 18–45 years); body mass = 82.6 ± 22.7 kg; stature = 1.78 ± 0.07 m) volunteered to participate in this study. Subjects had 5.0 ± 4.7 years of resistance-training experience and were familiarised with the PBP exercise. Their one-repetition maximum (1 RM) for the concentric-only and eccentric–concentric PBP variants were 88.8 ± 13.1 kg

(relative strength ratio = $1.11 \pm 0.19$) and $86.9 \pm 13.3$ kg (relative strength ratio = $0.96 \pm 0.23$), respectively. No physical limitations, health problems, or musculoskeletal injuries that could affect testing were reported. None of the subjects were taking drugs, medications, or dietary supplements known to influence physical performance. All subjects were informed of the study procedures and signed a written informed consent form before initiating the study. The study protocol adhered to the tenets of the Declaration of Helsinki and was approved by the Institutional Review Board.

### 2.2. Design

A repeated-measures design was used to compare the L-V relationship variables between the multiple-point and two-point methods and between the concentric-only and eccentric–concentric PBP variants. After a preliminary 1 RM testing session, subjects attended the faculty research laboratory on two separate occasions. A single PBP variant (concentric-only PBP or eccentric–concentric PBP) was evaluated in each session in a randomised order. The L-V relationship variables (i.e., $L_0$, $v_0$, and $A_{line}$) were obtained during each PBP variant using the multiple-point (four loads applied) and two-point (only the two most distant loads applied) methods. A linear velocity transducer (T-Force System version 3.70; Ergotech, Murcia, Spain) was used to collect the mean velocity (i.e., the average velocity from the first positive velocity until the velocity equalled 0 m·s$^{-1}$) of all repetitions throughout the study. The cable of the linear velocity transducer was attached vertically to the right side of the barbell using a velcro strap. The T-Force System interfaced to a personal computer by means of a 14-bit resolution analog-to-digital data acquisition board and custom software. Instantaneous velocity was sampled at a frequency of 1000 Hz and subsequently smoothed with a 4th order low-pass Butterworth digital filter with no phase shift and a 10 Hz cut-off frequency. The validity and reliability of the T-Force System for the recording of mean velocity have been reported elsewhere [21]. Subjects were forbidden from engaging in any vigorous activity over the course of the study. For each subject, all sessions were spaced 48–72 h apart and took place at the same time of day ($\pm 1$ h) to reduce the impact of the circadian rhythm on physical performance.

### 2.3. Procedures

A preliminary session was used to determine the 1 RM in the concentric-only PBP variant through a standard incremental loading test [22]. Body mass (Tanita BC 418 segmental, Tokyo, Japan) and stature (Seca 202 Stadiometer, Seca Ltd., Hamburg, Germany) were measured at the beginning of the first session. The warm-up consisted of jogging, dynamic stretching, and upper-body joint mobilisation exercises, followed by two sets of five repetitions of the concentric-only PBP variant against 20 and 30 kg. The initial external load of the incremental loading test was set at 40 kg and was progressively increased in 10 kg increments until the mean velocity was lower than 0.80 m·s$^{-1}$. Thereafter, the load was increased by five to one kg until the 1 RM was directly determined. Two repetitions were performed with light–moderate loads (mean velocity $\geq 0.80$ m·s$^{-1}$) and one repetition with heavier loads (mean velocity $< 0.80$ m·s$^{-1}$). Recovery time was set to three minutes for light–moderate loads (i.e., those $> 0.80$ m·s$^{-1}$) and five minutes for heavier loads (i.e., those $\geq 0.80$ m·s$^{-1}$). Subjects verbally received mean velocity performance feedback immediately after completing each repetition to encourage maximal effort.

The general warm-up specified for the preparatory session was used to begin each experimental session. The specific warm-up consisted of one set of ten, five, and two repetitions at 40, 60, and 80% of 1 RM, determined during the preliminary session, respectively. In a sequential order, subjects then performed three repetitions against the 14 kg load (L1; the lightest load), two repetitions against 85% of the 1 RM load (L4 = $72.0 \pm 10.9$ kg; the heaviest load), three repetitions against an equidistant intermediate light load (L2 = $33.2 \pm 3.6$ kg), two repetitions against an equidistant intermediate heavy load (L3 = $51.9 \pm 7.5$ kg), and one to five 1 RM attempts ($2.3 \pm 1.9$). The inter-repetition rest was

set at ten seconds and the inter-set rest was fixed at three minutes for submaximal loads and at five minutes for 1 RM attempts.

The PBP was performed in a Smith machine (Multipower Fitness Line, Peroga, Murcia, Spain) using bumper-calibrated plates (RusterFitness, Jaén, Spain). The PBP technique involved the subjects lying down in a prone position with the chin in contact with the bench and the legs held with a rigid strap on the calves [17]. There was 11 cm between the underside of the bench and the subjects' chests. The repetition was deemed invalid when the barbell failed to contact the bench's underside. To guarantee safety, there were two spotters on either side of the barbell. The two PBP variants' distinct characteristics are outlined below.

*Concentric-only PBP.* The assignment was started with the subjects holding the barbell with their elbows completely extended using a self-selected grip breadth. Note that the barbell was supported by the two telescopic holders of the Smith machine. From that position, subjects were instructed to pull the barbell as fast as possible until it made contact with the underside of the bench. The range of motion was individually measured and kept constant throughout all sessions using the two telescopic holders of the Smith machine.

*Eccentric–concentric PBP.* With the assistance of two spotters, subjects started the assignment by holding the barbell at a self-determined grip width and flexing their elbows to keep the barbell in contact with the bench's underside. From that position, subjects lowered the barbell in a controlled manner until contact was made with the two telescopic holders of the Smith machine. At that point, they pulled the barbell as fast as possible until it made contact with the underside of the bench.

### 2.4. L-V Relationship Modelling

The L-V relationships were modelled using the data points acquired from the four loads (i.e., multiple-point (L1-L2-L3-L4)) or the two most distant loads (i.e., two-point (L1-L4)) (Figure 1). A least-squares linear regression model $L(V) = L_0 - s \times V$ was used to determine the individualised L-V relationships, where $L_0$ represents the theoretical load at 0 velocity and $s$ is the slope of the L-V relationship [4,5]. The $v_0$ and $A_{line}$ were then calculated as follows: $v_0 = L_0/s$ and $A_{line} = L_0 \times v_0/2$ [4]. Only the repetition with the highest mean velocity value at each load was used for modelling the L-V relationships [7,23].

### 2.5. Statistical Analyses

Descriptive data are presented as means, SD, and range. The normal distribution of the data was confirmed using the Shapiro–Wilk test ($p > 0.05$). The strength of the L-V relationships modelled by the multiple-point method was examined through the Pearson's multivariate coefficients of determination ($r^2$). A two-way repeated-measures analysis of variance (ANOVA) (method (multiple-point vs. two-point) × PBP variant (concentric-only vs. eccentric–concentric)) with Bonferroni post hoc tests was applied to each L-V relationship variable. The magnitude of the differences was assessed with Cohen's $d$ effect size (ES), which was interpreted using the following scale: *trivial* (<0.20), *small* (0.20–0.59), *moderate* (0.60–1.19), *large* (1.20–2.00), and *very large* (2.00) [24]. The association of the L-V relationship variables between both methods and PBP variants was assessed by the Pearson's product-moment correlation coefficients ($r$). The strength of the $r$ coefficients was interpreted as *trivial* (0.00–0.09), *small* (0.10–0.29), *moderate* (0.30–0.49), *large* (0.50–0.69), *very large* (0.70–0.89), *nearly perfect* (0.90–0.99), or *perfect* (1.00) [24]. Statistical analyses were performed using the software package SPSS (IBM SPSS version 25.0, Chicago, IL, USA). Alpha was set at a $p \leq 0.05$ level.

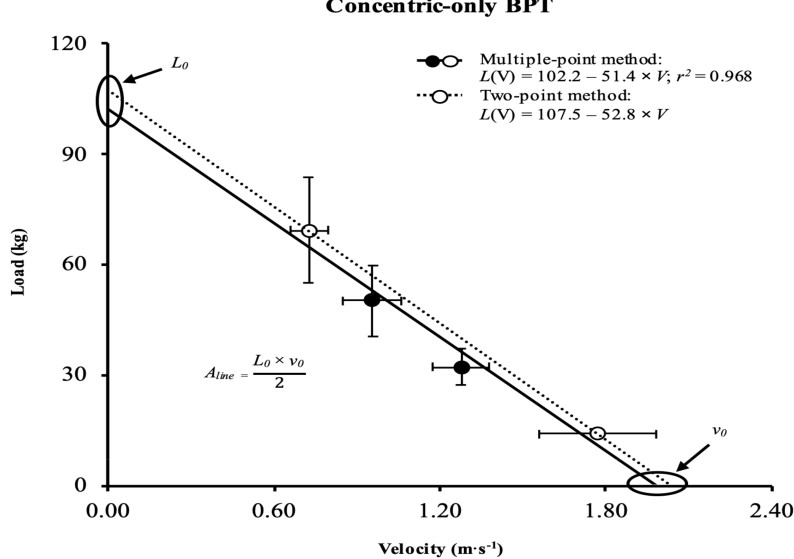

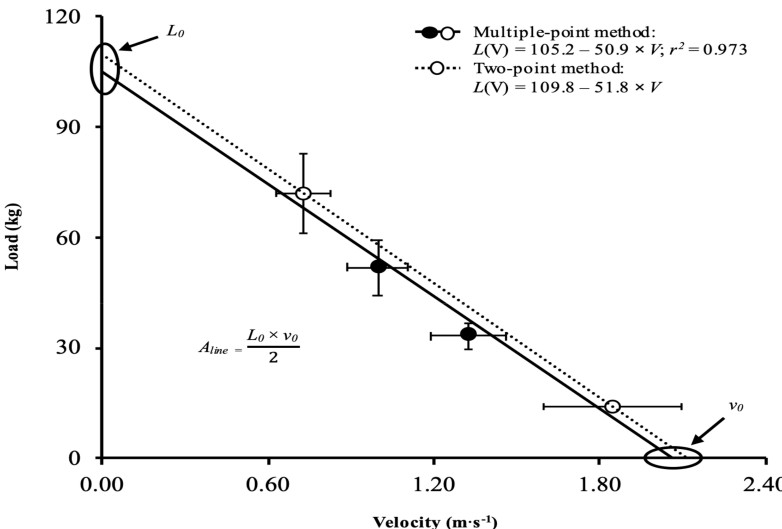

**Figure 1.** Load–velocity relationship obtained from the data averaged across the subjects modelled by the multiple-point (filled circles and continuous line) and two-point (open circles and dotted line) methods during the concentric-only (upper panel) and eccentric–concentric (lower panel) prone bench pull (PBP) variants. The regression equations and Pearson's multivariate coefficients of determination ($r^2$) are depicted. $L_0$, load–axis intercept; $v_0$, velocity–axis intercept; $A_{line}$, area under the load–velocity relationship line.

## 3. Results

The individualised L-V relationships obtained from the multiple-point method were strong for the concentric-only ($r^2$ range = 0.889–0.998) and eccentric–concentric ($r^2$ range = 0.889–0.992) PBP variants.

The ANOVA conducted on the L-V relationship variables did not reveal any significant method × PBP variant interaction ($F_{(1,22)} \leq 1.2$; $p \geq 0.281$). A significant main effect of method ($F_{(1,22)} \geq 79.3$; $p < 0.001$) but not of variant ($F_{(1,22)} \leq 2.2$; $p \geq 0.148$) was observed for the three L-V relationship variables (Table 1). Specifically, the two-point method provided significantly higher values for the three L-V relationship variables compared with the multiple-point method ($L_0$: 5.5 kg [95% confidence interval (CI) = 4.3 to 6.7 kg], $p < 0.001$, ES = 0.17; $v_0$: 0.05 m·s$^{-1}$ [95% CI = 0.04 to 0.06 m·s$^{-1}$], $p < 0.001$, ES = 0.20; $A_{line}$: 8.3 kg·m·s$^{-1}$ [95% CI = 6.6 to 10.0], $p < 0.001$, ES = 0.43).

Nearly perfect correlations ($r$ range = 0.972–0.995) were observed between the multiple-point and two-point methods for both PBP variants (Figure 2). Very large to nearly perfect correlations ($r$ range = 0.798–0.909) were observed between the concentric-only and eccentric–concentric PBP variants for both methods (Figure 3).

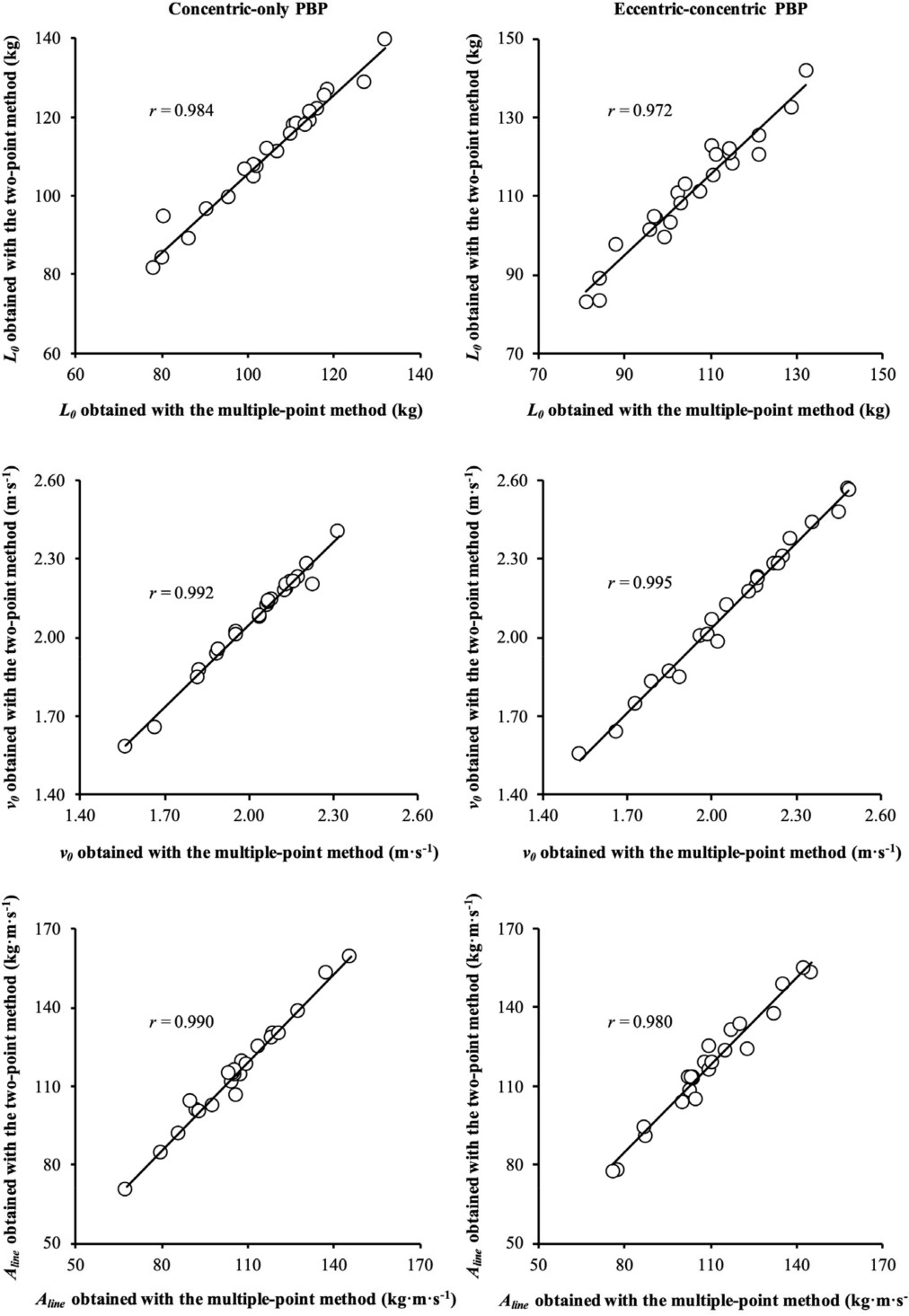

**Figure 2.** Correlations between the load–velocity relationship variables obtained with the multiple-point and two-point methods during the concentric-only (left panels) and eccentric–concentric (right panels) prone bench pull (PBP) variants. $L_0$, load–axis intercept; $v_0$, velocity–axis intercept; $A_{line}$, area under the load–velocity relationship line; $r$, Pearson's product-moment correlation coefficient.



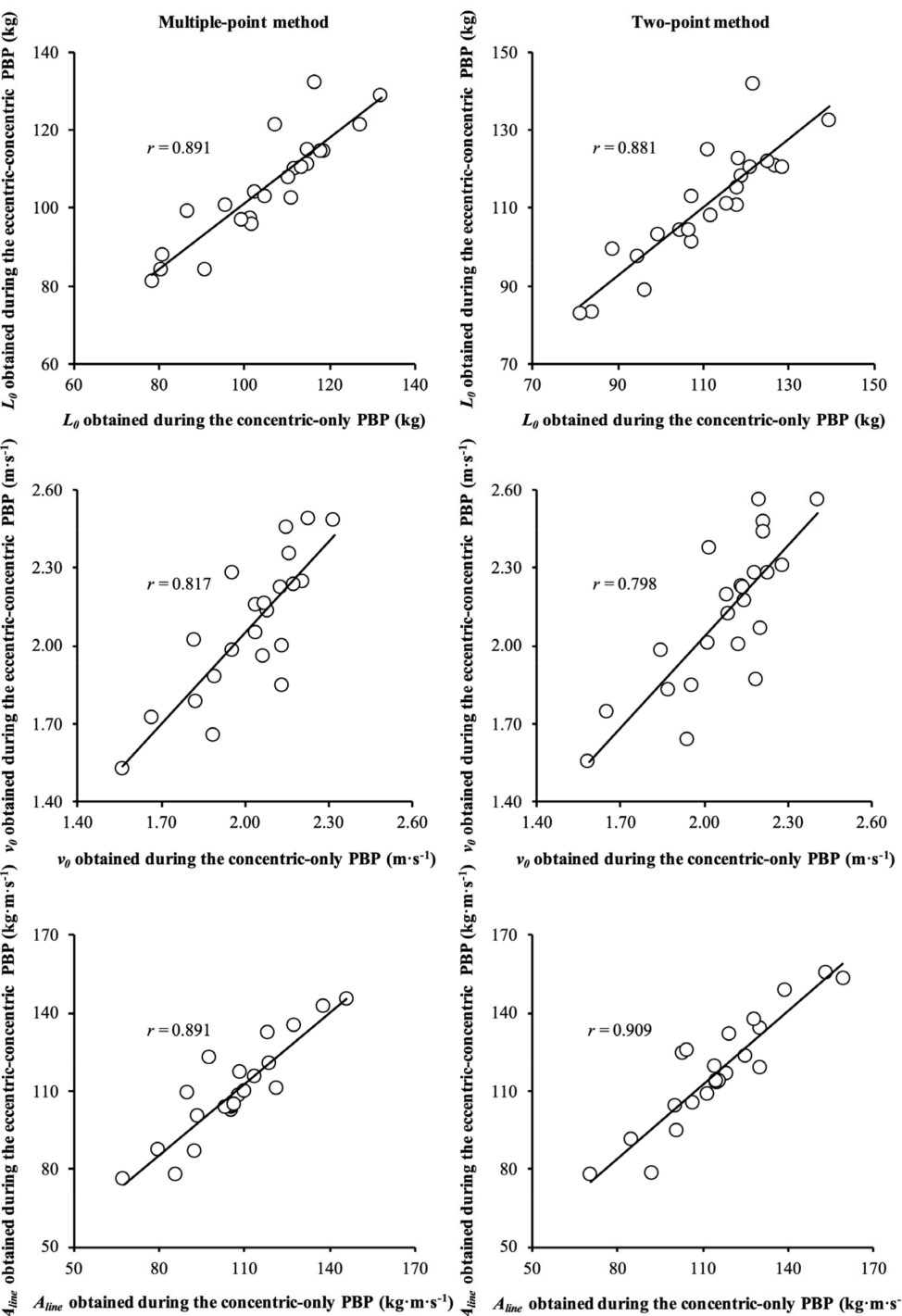

**Figure 3.** Correlations between the load–velocity relationship variables obtained during the concentric-only and eccentric–concentric prone bench pull (PBP) variants from the multiple-point (left panels) and two-point (right panels) methods. $L_0$, load–axis intercept; $v_0$, velocity–axis intercept; $A_{line}$, area under the load–velocity relationship line; $r$, Pearson's product-moment correlation coefficient.

**Table 1.** Two-way repeated-measures analysis of variance (ANOVA) comparing the load–velocity (L-V) relationship variables between the methods and prone bench pull (PBP) variants.

| L-V Relationship Variable | Method | PBP Variant | | ANOVA | |
|---|---|---|---|---|---|
| | | Concentric-Only | Eccentric–Concentric | Main Effects | Interaction |
| $L_0$ (kg) | Multiple-point | $105 \pm 15$ [78–132] | $105 \pm 14$ [81–132] | *Method:* $F_{(1,22)} = 89.4$; $p < 0.001$ *PBP variant:* $F_{(1,22)} = 0.2$; $p = 0.894$ | *Method $\times$ PBP variant:* $F_{(1,22)} = 0.3$; $p = 0.579$ |
| | Two-point | $111 \pm 15$ [81–140] | $111 \pm 15$ [83–142] | | |
| $v_0$ (m·s$^{-1}$) | Multiple-point | $2.03 \pm 0.18$ [1.56–2.32] | $2.08 \pm 0.26$ [1.53–2.49] | *Method:* $F_{(1,22)} = 79.3$; $p < 0.001$ *PBP variant:* $F_{(1,22)} = 2.2$; $p = 0.148$ | *Method $\times$ PBP variant:* $F_{(1,22)} = 0.5$; $p = 0.501$ |
| | Two-point | $2.07 \pm 0.19$ [1.59–2.41] | $2.12 \pm 0.28$ [1.56–2.57] | | |
| $A_{line}$ (kg·m·s$^{-1}$) | Multiple-point | $106 \pm 18$ [68–146] | $109 \pm 18$ [76–145] | *Method:* $F_{(1,22)} = 103.5$; $p < 0.001$ *PBP variant:* $F_{(1,22)} = 2.1$; $p = 0.157$ | *Method $\times$ PBP variant:* $F_{(1,22)} = 1.2$; $p = 0.281$ |
| | Two-point | $115 \pm 20$ [70–159] | $117 \pm 21$ [77–155] | | |

Descriptive values are presented as mean $\pm$ standard deviation [range]. $L_0$, load–axis intercept; $v_0$, velocity–axis intercept; $A_{line}$, area under the L-V relationship line; F, Snedecor's F; $p$, $p$-value.

## 4. Discussion

This study tested the hypothesis, in two variants of the PBP exercise, that lifting more than two loads (multiple-point method) in the testing procedure of the L-V relationship would cause fatigue, manifested in a lower magnitude of the L-V relationship variables in comparison with lifting only two loads (two-point method). The main findings of the present study revealed that the L-V relationship variables were as follows: (i) slightly greater for the two-point method compared with the multiple-point method, (ii) comparable between the concentric-only and eccentric–concentric PBP variants, and (iii) highly correlated between both methods and PBP variants. These results suggest that, regardless of the PBP variant, the testing procedure of the L-V relationship variables should be based on lifting only two loads because the addition of intermediate loads slightly decreases their magnitude.

Supporting our first hypothesis, the two-point method reported significantly higher L-V relationship variables than the multiple-point method. These results are in contrast with Pérez-Castilla et al. [5], who observed comparable L-V relationship variables between the multiple-point and two-point methods obtained during countermovement-jump exercise when a heavy squat load was included or excluded from the modelling ($p \geq 0.168$; ES $\leq 0.04$). Notably, the lack of statistical significance could be owing to the two-point method data being extracted from the multiple-point testing procedure. However, our results are in agreement with Garcia-Ramos et al. [12] who, during leg cycling exercise, observed a significantly lower maximum power for the multiple-point method compared with the two-point method applied in field conditions ($p = 0.041$; ES $= 0.36$) due to a decrease in maximum force ($p = 0.039$; ES $= 0.41$) but not in $v_0$ ($p = 0.570$; ES $= -0.15$). While Garcia-Ramos et al. [12] evaluated the multiple-point and two-point methods following an incremental loading test on separate days, in the present study two intermediate loads were measured after the two distant loads to evaluate both methods within the same session. It seems logical to speculate that the three L-V relationship variables were compromised in the present study by fatigue induced during the standard testing procedure based on multiple loads [12,25]. The L-V relationship in the PBP exercise was used in the present study to provide novel evidence supporting the two-point method as a simpler, quicker, and less fatigue-inducing strategy for determining maximum neuro-muscular capacities.

Our second hypothesis was also rejected since the L-V relationship variables were comparable between the concentric-only and eccentric–concentric PBP variants. These results contradict those of García-Ramos et al. [20], who reported a steeper L-V relationship (i.e., higher velocities across the full range of loads) for the eccentric–concentric bench

press variant compared with the concentric-only bench press variant due to the use of the stretch–shortening cycle. However, it is important to note that the barbell kinematics and kinetics greatly differ between the bench press and PBP exercises because of the different architecture of the muscles involved in these tasks [13]. In addition, during the PBP exercise the barbell is pulled towards the chest from a prone horizontal position (shoulder extension and elbow flexion), while the bench press exercise is based on a pushing action from a supine horizontal position (shoulder flexion and elbow extension) [13]. In each case, although no significant differences were observed between the PBP variants, the $v_0$ and $A_{line}$ were slightly higher for the eccentric–concentric PBP variant than for the concentric-only PBP variant. The use of a slight/moderate eccentric overload may likely improve force production at the beginning of the pulling action by reducing muscle slack [19]. This advantage may be compromised by the use of a heavy eccentric overload where the subjects have higher deceleration demands to brake the barbell over the target distance. Altogether, the magnitudes of the L-V relationship variables are comparable between the concentric-only and eccentric–concentric PBP variants. Future studies should compare the kinematic and kinetic characteristics between both PBP variants.

Supporting our third hypothesis, the L-V relationship variables were highly correlated between methods and PBP variants. These results are in line with Pérez-Castilla et al. [5] who reported a strong relationship between the multiple-point and two-point methods for the L-V relationship variables obtained during countermovement-jump exercise ($r \geq 0.96$). Similarly, our results are also in consensus with García-Ramos et al. [20] who revealed that mean test velocity (i.e., the average mean velocity value obtained across the whole L-V spectrum) was significantly and positively correlated between the concentric-only and eccentric–concentric bench press variants. Based on these findings, the L-V relationship variables obtained from both methods and PBP variants provide similar insight into maximal neuromuscular capacities. Practically, we recommend that practitioners use the two-point method (as it is easier, faster, and less likely to cause fatigue) and the concentric-only PBP variant (where no assistance is required to hold the bar during execution).

Despite the aforementioned results, readers should be mindful of a couple of limitations. Firstly, the same absolute loads were used in both exercises for modelling the L-V relationships based on the preliminary 1 RM obtained during the concentric-only PBP variant. However, since small differences in the 1 RM value were reported between the concentric-only and eccentric–concentric PBP variants (~2.7%), the relative loads (% 1 RM) were comparable for both PBP variants. Secondly, it was shown that the accuracy of the force–velocity relationship parameters not only depend on the equipment used (Smith machine vs. free-weight) [26], but also on the intrinsic characteristics of the experimental points used in the modelling (e.g., distance with respect to the intercept axes and their reliability) [27,28]. Therefore, caution should be taken when generalising the results of the present study to other exercises or testing conditions.

## 5. Conclusions

The two-point method provided higher L-V relationship variables than the multiple-point method, while the L-V relationship variables were comparable between the concentric-only and eccentric–concentric PBP variants. In addition, the L-V relationship variables were highly correlated between both methods and PBP variants and, therefore, they provide similar insight into maximal neuromuscular capacities. Practitioners are advised to use the two-point method as a simpler, quicker, and less fatigue-inducing procedure than the multiple-point method. From a logistical point of view, the concentric-only PBP variant is recommended over the eccentric–concentric PBP variant because it does not require the user to have assistance when holding the barbell during execution.

**Author Contributions:** Conceptualisation, A.G.-R. and A.P.-C.; methodology, S.M.-M., A.G.-R. and A.P.-C.; formal analysis, A.P.-C.; data curation, S.M.-M. and A.P.-C.; writing—original draft preparation, A.P.-C.; writing—review and editing, S.M.-M., A.G.-R., J.F.T.F. and A.P.-C.; visualisation, S.M.-M.,

A.G.-R., J.F.T.F. and A.P.-C.; supervision, A.P.-C. All authors have read and agreed to the published version of the manuscript.

**Funding:** This study was supported by the Spanish Ministry of University under a predoctoral grant (FPU19/01137).

**Institutional Review Board Statement:** The study was conducted in accordance with the Declaration of Helsinki, and approved by the Institutional Review Board of University of Granada (IRB approval: 2046/CEIH/2021).

**Informed Consent Statement:** Informed consent was obtained from all subjects involved in the study.

**Data Availability Statement:** Not applicable.

**Acknowledgments:** We would like to thank all the subjects who selflessly participated in the study.

**Conflicts of Interest:** The authors declare no conflict of interest.

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
