# Peer review of "Lifting More Than Two Loads Compromises the Magnitude of the Load–Velocity Relationship Variables: Evidence in Two Variants of the Prone Bench Pull Exercise"

_applsci, doi:10.3390/app13031944_

Round 1

Reviewer 1 Report

Firstly, I would like to congratulate to the authors of the manuscript. This study is easy to follow and well documented. However, as an external reviewer, I must do some comments or suggestions to increment the quality of the manuscript. Having said that, please see all my suggestions from a constructive point of view. This review report is divided in two parts: i.e., general comments as well as, specific comments. Please, feel free to discuss whatever point that you want to discuss. 

General comments

Page 2, lines 86 to 88. The authors stated as a research objective: “…the present study´s primary objective was to compare and associate the magnitude of the L-V relationship variables between the multiple-point and two-point methods and between the concentric-only and eccentric-concentric PBP variants…”. From my mind, the authors have introduced two research objectives in one. Let me explain deeply. I understand that one of the main goals of the study was compare the outcomes of interest (i.e., L0, Vand, Aline) between the two proposed protocols (i.e., multiple-point and two-point methods) and, between the two PBP variants (i.e., concentric-only, and eccentric-concentric). Due to the authors employed the same sample in all variants and protocols, this corresponds to a repeated measures design. Briefly, there are two independent variables with a total of 2 levels each (i.e., protocol and PBP variants). As the authors explained in statistical analysis section, this research objective is answered by a repeated-measures ANOVA. However, at this point it is important to differentiate the influence or the effect of these two independent variables on the measured outcomes and the relationship between the outcome of interest across the two protocols and variants done. Therefore, my suggestion is about to add a new research objective regarding the correlations. Consider that each research objective should be associate with a unique statistical technique.

Page 3, lines 99 – 101. I don´t fully understand the process employed to get the experimental data. Since the authors have said that there are two independent variables (i.e., method and PBP variants) that means that, if participants attended a couple of times to laboratory, they were assessed of the two independent variables in the same day, aren´t they?. For instance, Day 1 = 1RM protocol, Day 2 = method 1  [multiple-point method] and method 2 [two-point method] for PBP variant 1, and Day 3 = method 1 [multiple-point method] and method 2 [two-point method] for PBP variant 2, does it correct?. Please, if I am not correct, explain, in a better way, how the data was collected. Even more, what did you randomized? The starting order at each day and/or the days completed by participants?

Page 8, Figure 2 and 3. Please, add letter (i.e., A, B, C … and so on), to the figures displayed. In addition, I suggest to the authors add a double line at each axis to reflect that the axis don´t start at “0”. 

Specific comments 

(Page 1, lines 18-19). Please, add to your research objective the population of interest. Do the same in the last paragraph of the introduction section. 

Page 2, lines 96 – 97. As I understand, this study is not a crossover design. In any case, a repeated measures design. 

Page 4, lines 170 – 171. The authors said: “Only the repetition with the highest mean velocity value at each load was used  for modelling the L-V relationship”. A recent study published by Bautista et al. (2022) showed that differences in mean velocity could be obtained when L-V relationship was calculated using the best value vs. average value if the coefficient of variation was higher than 10%. The references employed by authors is referred to a back-squat exercise. (“A Load-velocity relationship variables to assess the maximal neuromuscular capacities during the back-squat exercise”). Please, consider add the reference because fit better with the proposed exercise employed. (REF: https://doi.org/10.1177/17479541211035499).

Page 5, lines 201 – 203. This is a cornerstone point in the manuscript. Firstly, there were a total of three outcomes (i.e., L0, Vand, Aline) analyzed. Although the authors said: “Specifically, the two-point method provided significantly higher values for the three L-V relationship variables compared to the multiple-point method ..”. My main question here is: why a better value in these outcomes means a better performance? From my actual knowledge, both calculations methods were obtained from an indirect calculation of “real” performance (understanding that “real” performance was the 1RM protocol done in the first assessment day). Therefore, my suggestion is to calculate the difference between the actual 1RM (day 1) vs. Land compare the differences obtained between the two proposed methods. Moreover, please, add the mean differences (MD) and 95% of confident intervals (95% CI)  in addition to p-value and effect size. Finally, although the authors declared statistically significant differences in method variable, the effect sizes obtained were range from trivial to small. 

Page 9, lines 243 to 246. The authors concluded: “these results suggest that the testing procedure of the L-V relationship variables should be based on lifting only two loads because the addition of intermediate loads decreases their magnitude, while the concentric-only PBP variant could be recommended because it is simpler as it does not require external help to hold the barbell.”. Let me split these points: 

Point one, I am not sure about if with the results obtained the authors are in disposition to make this affirmation (“results suggest that the testing procedure of the L-V relationship variables should be based on lifting only two loads because the addition of intermediate loads decreases their magnitude”). How is explained before, both methods employed are estimations of the “real” performance. Now, it is important to know what method reflected in a better way the real performance. On the other hand, if two-point method reflected a better performance, the magnitude of differences (effect size) between methods were categorized from trivial to small.

Point two, regarding the scope of this study, that is, an ACUTE effect of calculation method on L-V relationship, there is not possible to extrapolate or assume that these differences would be found if an intervention study is done. I mean, if I do a study where after an intervention period I use the L-V relationship (obtained from two-point or multiple-point method), which calculation method will reflect better the performance obtained? Or even more, does the fatigue the responsible of the differences across methods? 

Finally, the sentence: “while the concentric-only PBP variant could be recommended because it is simpler as it does not require external help to hold the barbell” … is based on practical applications rather than results obtained. Again, the results obtained showed no differences in PBP variant nor Method x PBP variant. Although I am not telling that the previous sentence is not true or adequate, I am only saying that is not supported by data obtained. I guess that this sentence make sense in “practical application section”.  

Therefore, I suggest to the authors re-write this sentence to synthesized, with a more precise words, the results obtained. For example, “ … our results suggested that with two-point method testing procedure the L-V relationship variables (i.e., L0, Vand, Aline) were statistically different, from trivial to small effect size, than the results obtained from multiple-point method…”. 

Author Response

General comments

Firstly, I would like to congratulate to the authors of the manuscript. This study is easy to follow and well documented. However, as an external reviewer, I must do some comments or suggestions to increment the quality of the manuscript. Having said that, please see all my suggestions from a constructive point of view. This review report is divided in two parts: i.e., general comments as well as, specific comments. Please, feel free to discuss whatever point that you want to discuss.

Response

The reviewer’s comments are highly appreciated. We have considered all suggestions and we believe our manuscript is stronger as a result of the changes that we have introduced in the revised version of the manuscript. An itemized point-by-point response to the reviewer’s comments is presented below:

Comment

Page 2, lines 86 to 88. The authors stated as a research objective: “…the present study´s primary objective was to compare and associate the magnitude of the L-V relationship variables between the multiple-point and two-point methods and between the concentric-only and eccentric-concentric PBP variants…”. From my mind, the authors have introduced two research objectives in one. Let me explain deeply. I understand that one of the main goals of the study was compare the outcomes of interest (i.e., L0, V0 and, Aline) between the two proposed protocols (i.e., multiple-point and two-point methods) and, between the two PBP variants (i.e., concentric-only, and eccentric-concentric). Due to the authors employed the same sample in all variants and protocols, this corresponds to a repeated measures design. Briefly, there are two independent variables with a total of 2 levels each (i.e., protocol and PBP variants). As the authors explained in statistical analysis section, this research objective is answered by a repeated-measures ANOVA. However, at this point it is important to differentiate the influence or the effect of these two independent variables on the measured outcomes and the relationship between the outcome of interest across the two protocols and variants done. Therefore, my suggestion is about to add a new research objective regarding the correlations. Consider that each research objective should be associate with a unique statistical technique.

Response

We fully agree with the reviewer’s comment. This information has been rewritten in the revised version of the manuscript and now it reads: “Specifically, the aims of the study were (i) to compare and (ii) to associate the magnitude of the L-V relationship variables between the multiple-point and two-point methods and between the concentric-only and eccentric-concentric PBP variants in resistance-trained males.”

Comment

Page 3, lines 99 – 101. I don´t fully understand the process employed to get the experimental data. Since the authors have said that there are two independent variables (i.e., method and PBP variants) that means that, if participants attended a couple of times to laboratory, they were assessed of the two independent variables in the same day, aren´t they?. For instance, Day 1 = 1RM protocol, Day 2 = method 1  [multiple-point method] and method 2 [two-point method] for PBP variant 1, and Day 3 = method 1 [multiple-point method] and method 2 [two-point method] for PBP variant 2, does it correct?. Please, if I am not correct, explain, in a better way, how the data was collected. Even more, what did you randomized? The starting order at each day and/or the days completed by participants?

Response

Subjects attended the laboratory on two separate occasions. As we have indicated, a single prone bench pull variant (concentric-only or eccentric-concentric) was evaluated in each session in a randomised order. Specifically, the load-velocity relationship variables were obtained during each prone bench pull variant using the multiple-point (four loads applied) and two-point (only the two most distant loads applied) methods. This information has been clarified in the revised version of the manuscript.

Comment

Page 8, Figure 2 and 3. Please, add letter (i.e., A, B, C … and so on), to the figures displayed. In addition, I suggest to the authors add a double line at each axis to reflect that the axis don´t start at “0”.

Response

We have decided to modify the Figure to facilitate the understanding of non-expert readers. Specifically, each prone bench pull variant with the two methods (multiple-point and two-point) is presented in separate panels.

Comment

Page 1, lines 18-19. Please, add to your research objective the population of interest. Do the same in the last paragraph of the introduction section.

Response

This information has been provided in last paragraph of the introduction section. However, it is already indicated in the abstract that the population of interest were resistance-trained males.

Comment

Page 2, lines 96 – 97. As I understand, this study is not a crossover design. In any case, a repeated measures design.

Response

We agree with the reviewer’s comment. This information has been rewritten in the revised version of the manuscript and now it reads: “A repeated-measures design was used to compare the L-V relationship variables between the multiple-point and two-point methods and between the concentric-only and eccentric-concentric PBP variants.”

Comment

Page 4, lines 170 – 171. The authors said: “Only the repetition with the highest mean velocity value at each load was used for modelling the L-V relationship”. A recent study published by Bautista et al. (2022) showed that differences in mean velocity could be obtained when L-V relationship was calculated using the best value vs. average value if the coefficient of variation was higher than 10%. The references employed by authors is referred to a back-squat exercise. (“A Load-velocity relationship variables to assess the maximal neuromuscular capacities during the back-squat exercise”). Please, consider add the reference because fit better with the proposed exercise employed. (REF: https://doi.org/10.1177/17479541211035499).

Response

The reference of Bautista et al. (2022) has been added to the revised version of the manuscript.

Comment

Page 5, lines 201 – 203. This is a cornerstone point in the manuscript. Firstly, there were a total of three outcomes (i.e., L0, V0 and, Aline) analyzed. Although the authors said: “Specifically, the two-point method provided significantly higher values for the three L-V relationship variables compared to the multiple-point method ..”. My main question here is: why a better value in these outcomes means a better performance? From my actual knowledge, both calculations methods were obtained from an indirect calculation of “real” performance (understanding that “real” performance was the 1RM protocol done in the first assessment day). Therefore, my suggestion is to calculate the difference between the actual 1RM (day 1) vs. L0 and compare the differences obtained between the two proposed methods. Moreover, please, add the mean differences (MD) and 95% of confident intervals (95% CI) in addition to p-value and effect size. Finally, although the authors declared statistically significant differences in method variable, the effect sizes obtained were range from trivial to small.

Response

The objective of this study is not to estimate the 1RM from the L0 but to obtain the maximal neuromuscular capacities from a simpler and more precise alternative such as the load-velocity relationship (see Pérez-Castilla 2021 and 2022 for further details). Note that maximal neuromuscular capacities can be obtained by using two (i.e., two-point method) or more (i.e., multiple-point method) loading conditions in the load-velocity modeling (Pérez-Castilla et al. 2022). Therefore, we have only indicated that the two-point method reported higher values for the three load-velocity relationship variables compared to the multiple-point method (i.e., higher capacity to produce force at low [L0] and high [v0] velocities as well as doing work at a maximal rate [Aline]). We have provided the mean differences and 95% of confident intervals following the reviewer’s suggestions.

Pérez-Castilla, A.; Jukic, I.; García-Ramos, A. Validation of a novel method to assess maximal neuromuscular capacities through the load-velocity relationship. J. Biomech. 2021, 127, 110684. doi:10.1016/j.jbiomech.2021.110684

Pérez-Castilla, A.; Jukic, I.; Janicijevic, D.; Akyildiz, Z.; Senturk, D.; García-Ramos, A. Load-velocity relationship variables to assess the maximal neuromuscular capacities during the back-squat exercise. Sports Health. 2022, 14, 885-893. doi:10.1177/19417381211064603

Pérez-Castilla, A.; Ramirez-Campillo, R.; Fernandes, J. F. T.; García-Ramos, A. Feasibility of the 2-point method to determine the load−velocity relationship variables during the countermovement jump exercise. J. Sport Health Sci. 2021, online ahead of print. doi:10.1016/j.jshs.2021.11.003

Comment

Page 9, lines 243 to 246. The authors concluded: “these results suggest that the testing procedure of the L-V relationship variables should be based on lifting only two loads because the addition of intermediate loads decreases their magnitude, while the concentric-only PBP variant could be recommended because it is simpler as it does not require external help to hold the barbell.”. Let me split these points:

Point one, I am not sure about if with the results obtained the authors are in disposition to make this affirmation (“results suggest that the testing procedure of the L-V relationship variables should be based on lifting only two loads because the addition of intermediate loads decreases their magnitude”). How is explained before, both methods employed are estimations of the “real” performance. Now, it is important to know what method reflected in a better way the real performance. On the other hand, if two-point method reflected a better performance, the magnitude of differences (effect size) between methods were categorized from trivial to small.

Point two, regarding the scope of this study, that is, an ACUTE effect of calculation method on L-V relationship, there is not possible to extrapolate or assume that these differences would be found if an intervention study is done. I mean, if I do a study where after an intervention period I use the L-V relationship (obtained from two-point or multiple-point method), which calculation method will reflect better the performance obtained? Or even more, does the fatigue the responsible of the differences across methods?

Response

We would like to clarify that we have used the word suggest and not affirm for the reasons stated by the reviewer. In this sense, although the magnitude of the differences ranges from trivial to small, the three variables of the load-velocity relationship were always greater for the two-point method than for the multiple-point method. Based on previous studies (García-Ramos et al., 2018 y 2021), this fact could be attributed to the fatigue induced by the standard testing procedure based on multiple loads. Therefore, the two-point method is not only simpler and quicker, but also less prone to fatigue than the multiple-point method to determine the maximal neuromuscular capacities during the prone bench pull exercise. In any case, if the reviewer has a better proposal to interpret the results of the present study, we will be happy to include it in the next revision of the manuscript.

Garcia-Ramos, A.; Zivkovic, M.; Djuric, S.; Majstorovic, N.; Manovski, K.; Jaric, S. Assessment of the two-point method applied in field conditions for routine testing of muscle mechanical capacities in a leg cycle ergometer. Eur. J. Appl. Physiol. 2018, 118, 1877-1884. doi:10.1007/s00421-018-3925-9

García-Ramos, A.; Pérez-Castilla, A.; Jaric, S. Optimisation of applied loads when using the two-point method for assessing the force-velocity relationship during vertical jumps. Sports Biomech. 2021, 20, 274-289. doi:10.1080/14763141.2018.1545044

Comment

Finally, the sentence: “while the concentric-only PBP variant could be recommended because it is simpler as it does not require external help to hold the barbell” … is based on practical applications rather than results obtained. Again, the results obtained showed no differences in PBP variant nor Method x PBP variant. Although I am not telling that the previous sentence is not true or adequate, I am only saying that is not supported by data obtained. I guess that this sentence make sense in “practical application section”. 

Response

We fully agree with the reviewer’s comment. This information has been rewritten in the revised version of the manuscript and now it reads: “These results suggest that, regardless of the PBP variant, the testing procedure of the L-V relationship variables should be based on lifting only two loads because the addition of intermediate loads slightly decreases their magnitude.”

Comment

Page 9, lines 243 to 246. Therefore, I suggest to the authors re-write this sentence to synthesized, with a more precise words, the results obtained. For example, “ … our results suggested that with two-point method testing procedure the L-V relationship variables (i.e., L0, V0 and, Aline) were statistically different, from trivial to small effect size, than the results obtained from multiple-point method…”.

Response

We have changed “significantly greater” to “slightly greater” in the revised version of the manuscript.

Reviewer 2 Report

The introduction and framing of the problem under study seems justified.

The methodology seems to be adequate, although we suggest the "Subjects" could precced "Design".

The results seem to be presented in an understandable way, as well as the discussion.

The conclusions could point future studies to be undertaken.

Author Response

Comment
The introduction and framing of the problem under study seems justified. The methodology seems to be adequate, although we suggest the "Subjects" could precede "Design". The results seem to be presented in an understandable way, as well as the discussion. The conclusions could point future studies to be undertaken.

Response

The reviewer’s comment is highly appreciated. We have changed the order subsections following the reviewer’s recommendations. In addition, future lines of research have been indicated throughout the discussion section. We believe that the conclusions section would not be the best place in the manuscript to indicate this information.

Reviewer 3 Report

Your hypothesis is good : "We hypothesized that the L-V relationship variables would be (i) greater for the two point method compared to multiple-point method [12], (ii) greater for the eccentric-concentric PBP variant compared to the concentric-only PBP variant [19], and (iii) highly cor-related between methods and PBP variants [5,20].

That is the reason why there are numerous mobile application calculating the F-V relationship and you neglicted all the scientific publications of JP MORIN which validated the application allowing automatisation for caches. There is nothing new in you study which does not pratical tips for coaches and athletes .

see the JB MORIN for helping you to improve this paper and to have a real bibliography outside (in addition) your own work  

https://www.researchgate.net/profile/Jean-Benoit-Morin

Author Response

Comment
Your hypothesis is good: "We hypothesized that the L-V relationship variables would be (i) greater for the two point method compared to multiple-point method [12], (ii) greater for the eccentric-concentric PBP variant compared to the concentric-only PBP variant [19], and (iii) highly cor-related between methods and PBP variants [5,20]. That is the reason why there are numerous mobile application calculating the F-V relationship and you neglicted all the scientific publications of JB MORIN which validated the application allowing automatisation for caches. There is nothing new in you study which does not pratical tips for coaches and athletes. See the JB MORIN for helping you to improve this paper and to have a real bibliography outside (in addition) your own work: https://www.researchgate.net/profile/Jean-Benoit-Morin.

Response

Our research group has been investigating the testing procedures of the force-velocity relationship for the last five years. In this sense, we are perfectly familiar with the line of research of Professor JB Morin. However, the reliability and validity of the parameters derived from the force-velocity relationship remain questionable (Cuk et al., 2014; Feeney et al., 2016; Iglesias-Soler et al., 2019; Kotani et al., 2022). This is at least partially influenced by the extrapolation needed from the experimental points to v0 (García-Ramos et al., 2018). Indeed, force outputs are not necessary for modelling the load-velocity relationship to assess maximal neuromuscular capabilities (Pérez-Castilla et al. 2021), thereby reducing the amount of extrapolation (N vs. kg against velocity) to v0. Therefore, it is not surprising that the load-velocity relationship variables have not only provided high reliability, but also high concurrent validity with respect to other traditional tests (Pérez-Castilla et al. 2021 and 2022). Specifically, our results suggest that the testing procedure of the load-velocity relationship variables should be based on lifting only two loads because the addition of intermediate loads decreases their magnitude, while the concentric-only prone bench pull variant could be recommended because it is simpler as it does not require external help to hold the barbell. Therefore, in disagreement with the reviewer, the results of the present study are of great interest within the velocity-based resistance training literature.

Cuk, I.; Markovic, M.; Nedeljkovic, A.; et al. Force-velocity relationship of leg extensors obtained from loaded and unloaded vertical jumps. Eur. J. Appl. Physiol. 2014, 114, 1703-1714.

Feeney, D.; Stanhope, S.J.; Kaminski, T.W.; Machi, A.; Jaric, S. Loaded vertical jumping: Force-velocity relationship, work, and power. J. Appl. Biomech. 2016, 32, 120-127.

Iglesias-Soler, E.; Mayo, X.; Rial-Vazquez, J.; et al. Reliability of force-velocity parameters obtained from linear and curvilinear regressions for the bench press and squat exercises. J. Sports. Sci. 2019, 37, 2596-2603.

Kotani, Y.; Lake, J.; Guppy, S.N.; et al. Reliability of the squat jump force-velocity and load-velocity profiles. J. Strength. Cond. Res. 2022, 36, 3000-3007.

Garcia-Ramos, A.; Perez-Castilla, A.; Jaric, S. Optimisation of applied loads when using the two-point method for assessing the force-velocity relationship during vertical jumps. Sports Biomech. 2022, 20: 274-289.

Pérez-Castilla, A.; Jukic, I.; García-Ramos, A. Validation of a novel method to assess maximal neuromuscular capacities through the load-velocity relationship. J. Biomech. 2021, 127, 110684. doi:10.1016/j.jbiomech.2021.110684

Pérez-Castilla, A.; Jukic, I.; Janicijevic, D.; Akyildiz, Z.; Senturk, D.; García-Ramos, A. Load-velocity relationship variables to assess the maximal neuromuscular capacities during the back-squat exercise. Sports Health. 2022, 14, 885-893. doi:10.1177/19417381211064603

Round 2

Reviewer 3 Report

this article has interesting practical application based on enough good scientific baseline and protocole.